# Explaining Datasets in Words:
# Statistical Models with Natural Language Parameters

**Ruiqi Zhong**[*]     **Heng Wang**     **Dan Klein**     **Jacob Steinhardt**

## Abstract

To make sense of massive data, we often first fit simplified models and then interpret the parameters; for example, we cluster the text embeddings and then interpret the mean parameters of each cluster. However, these parameters are often high-dimensional and hard to interpret. To make model parameters directly interpretable, we introduce a family of statistical models—including clustering, time series, and classification models—parameterized by *natural language predicates*. For example, a cluster of text about COVID could be parameterized by the predicate "*discusses COVID*". To learn these statistical models effectively, we develop a model-agnostic algorithm that optimizes continuous relaxations of predicate parameters with gradient descent and discretizes them by prompting language models (LMs). Finally, we apply our framework to a wide range of problems: taxonomizing user chat dialogues, characterizing how they evolve across time, finding categories where one language model is better than the other, clustering math problems based on subareas, and explaining visual features in memorable images. Our framework is highly versatile, applicable to both textual and visual domains, can be easily steered to focus on specific properties (e.g. subareas), and explains sophisticated concepts that classical methods (e.g. n-gram analysis) struggle to produce.[2]

## 1   Introduction

To analyze massive datasets, we often fit simplified statistical models and interpret the learned parameters. For example, to categorize a set of user queries, we might cluster their embeddings, look at samples from each cluster, and hopefully each cluster corresponds to an explainable category, e.g. "*asks about COVID symptoms*" or "*discusses the U.S. Election*". Unfortunately, each cluster might contain an uninterpretable group of queries, thus failing to explain the categories.

Such a failure is not an isolated incident: many models explain datasets by learning high dimensional parameters, but these parameters might require significant human effort to interpret. For example, BERTopic [18] learns uninterpretable cluster centers over high-dimensional neural embeddings. LDA [7], Dynamic Topic Modeling [6] (time series), and Naive Bayes (classification) learn weights over a large set of words/phrases, which do not directly explain abstract concepts [9, 52, 63]. We want model parameters that are more interpretable, since explaining datasets is important in machine learning [60], business [4], political discussion [47], and science [17, 34].

To make model parameters directly interpretable, we introduce a family of models where some of their parameters are represented as natural language predicates, which are inherently interpretable. Our core insight is that we can use a predicate to extract a 0/1 feature by checking whether it is true on a sample. For instance, given the predicate $\phi = $ "*discusses the U.S. Election*", its denotation $[\![\phi]\!]$ is a binary function that evaluates to 1 on texts $x$ discussing the U.S. Election and 0 otherwise:

$$[\![\phi : \text{``discusses the U.S. Election''}]\!](x : \text{``Is Georgia a swinging state this year?''}) = 1.$$

---

[*]ruiqi-zhong@berkeley.edu, corresponding author. All authors affiliated with UC Berkeley.

[2]Our code and dataset are at `https://github.com/ruiqi-zhong/nlparam`

38th Conference on Neural Information Processing Systems (NeurIPS 2024).

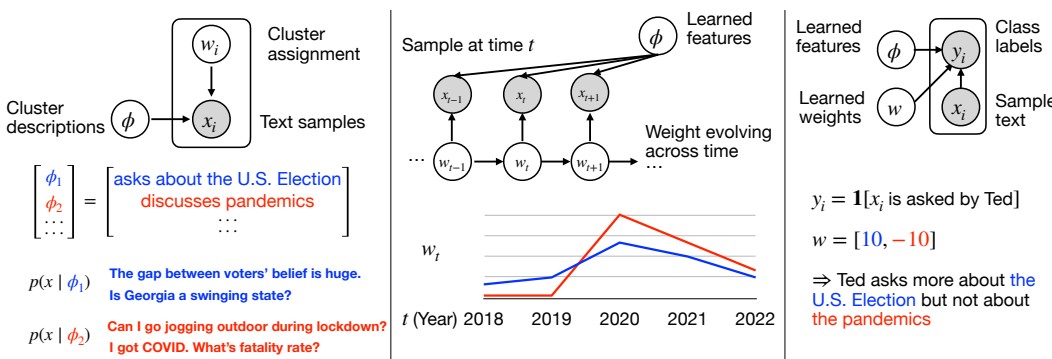

Figure 1: Our framework can use **natural language predicates** to parameterize a wide range of statistical models. **Left.** A clustering model that categorizes user queries. **Middle.** A time series model that characterizes how discussion changes across time. **Right.** A classification model that summarizes user traits. Once we define the model, we learn $\phi$ and $w$ based on $x$ (and $y$).

Using these 0/1 feature values, we define a wide variety of models, including clustering, classification, and time series modeling, all parameterized by natural langauage predicates (Figure 1).

Learning these predicates $\phi$ requires optimizing them to maximize the log-likelihood of the data. This is challenging because $\phi$ are discrete and thus do not admit gradient-based optimization. We propose a general method to effectively optimize $\phi$: we create a continuous relaxation $\tilde{\phi}$ of $\phi$ and optimize $\tilde{\phi}$ with gradient descent; then we prompt an LLM to explain the behavior of $\tilde{\phi}$, thus converting it back to discrete predicates (Section 4). We repeat this process to iteratively improve performance.

To evaluate our optimization algorithm, we create statistical modeling problems where the optimal predicate parameters are known, so we can use them as the ground truth. We evaluated on three different statistical models (clustering, multilabel classification, and time series modeling, as illustrated in Figure 1) and used five different datasets (NYT articles, AG-News, DBPedia, Bills, and Wiki [40, 58, 32, 23]). We found that both continuous relaxation and iterative refinement improve performance; additionally, our model-agnostic algorithm matches the performance (2% increase in F1 score) of the previous algorithm specialized for explainable text clustering [53].

Finally, we show that our framework is highly versatile by applying it to a wide range of tasks: taxonomizing user chat dialogues [59], characterizing how they evolve, finding categories where one language model is better than the other, clustering math problems [21] based on their subareas, and explaining what visual features make an image memorable [24]. Our framework applies to both text and visual domains, can be easily steered to explain specific abstract properties, and explains complicated concepts that classical methods (e.g. n-gram regression/topic model) struggle to produce. Combining LLM's ability to generate explanations along with traditional statistical models' ability to process sophisticated data patterns, our framework holds the promise to help humans better understand the complex world.

## 2 Related Work

**Statistical Modeling in Text.** Statistical models based on n-gram features or neural embeddings are broadly used to analyze text datasets. For example, logistic regression or naïve Bayes models are frequently used to explain differences between text distributions [51]; Gaussian mixture models on pre-trained embeddings can create text clusters [2]; topic models can mine major topics across a large collection of documents [7] and across time [6]. However, since these models usually rely on high-dimensional parameters, they are difficult to interpret: for example, human studies from [9] show that the most probable words for a topic might not form a semantically coherent category. To interpret these models, prior works proposed to explain each topic or cluster by extracting candidate phrases either from the corpus or from Wikipedia [8, 49, 57]. Our work complements these approaches to explain models with natural language predicates, which are potentially more flexible.

**Prompting Language Model to Explain Dataset Patterns.** Our algorithm heavily relies on the ability of LLMs to explain distributional patterns in data when prompted with datasets [39, 46]. [61, 62, 15] have prompted LLMs to explain differences between two text distributions; [53, 38, 50, 27] prompted LLMs to generate topic descriptions over unstructured texts; [44, 22, 64] prompted LLMs to explain the function that maps from an input to an output; [45, 5] prompted LLMs to explain what inputs activate a direction in the neural embedding space. However, these works focused on individual applications or models in isolation; in contrast, our work creates a unifying framework to define and learn more complex models (e.g. time series) with natural language parameters.

**Concept Bottleneck Models (CBM).** CBMs aim to achieve explainability by learning a simple model over a set of interpretable features [26], and recent works have proposed to extract these features using natural language phrases/predicates [3, 55, 30, 12, 41]. While most of these works focus on classification tasks, our work formalizes a broad family of models—including clustering and time series —and proposes a model-agnostic algorithm to learn them. Additionally, these prior works focus on downstream task performance (e.g. classification accuracy), thus implicitly assuming that the model grounds the feature explanations in the same way as humans; in contrast, since our focus is on explanations, we focus on our algorithm's ability to recover ground truth explainable features.

We discuss more related work on discrete prompt optimization, exploratory analysis, and learning with latent language in Appendix A.

## 3 Mathematical Formulation

### 3.1 Predicate-Conditioned Distribution

In order to model text distributions with natural language parameters, we introduce a new family of distributions, *predicate-conditioned distributions*; these distributions will serve as building blocks for the models introduced later, just like normal distributions are building blocks for many classical models like Gaussian Mixture or Kalman Filter. Predicate-conditioned distributions $p$ are supported on the set $X$ of all the text samples we observe from the dataset, and they are parameterized by (1) a list of $K$ predicates $\vec{\phi} \in \Phi^K$, and (2) real-valued weights $w \in \mathbb{R}^K$ on those predicates. Formally,

$$p(x \mid \vec{\phi}, w) \propto e^{w^T [\![\vec{\phi}]\!](x)}. \tag{1}$$

We now explain how to (1) extract a feature vector from $x$ using $\vec{\phi}$, (2) linearly combine $\vec{\phi}$ by re-weighting with $w$, and (3) use the reweighted values to define $p(x \mid w, \vec{\phi})$.

**Natural Language Parameters $\vec{\phi}$.** Each predicate $\phi \in \Phi$ is a natural language string and its denotation $[\![\phi]\!] : X \to \{0, 1\}$ maps samples to their value under the predicate. For example, if $\phi =$ "*is sports-related*", then $[\![\phi]\!]$("*I love soccer.*")$= 1$. Since a model typically requires multiple features to explain the data, we consider vectors $\vec{\phi} \in \Phi_K$ of $K$ predicates, where now $[\![\vec{\phi}]\!]$ maps $X$ to $\{0, 1\}^K$:

$$[\![\vec{\phi}]\!](x) := \big([\![\phi_1]\!](x), [\![\phi_2]\!](x), \ldots, [\![\phi_K]\!](x)\big). \tag{2}$$

To instantiate $[\![\cdot]\!]$ computationally, we prompt a language model to check whether $\phi$ is true on the input $x$, following the practice from prior works [61, 62]. See Figure 2 (left) for the prompt we used.

**Reweighting with $w$.** Consider the following example:

$$w = [-5, 3]; \quad \vec{\phi} = [\text{"is in English", "is sports-related"}]. \tag{3}$$

Then $w^T [\![\vec{\phi}]\!]$ has a value of $-5 \cdot 1 + 3 \cdot 0 = -5$ for an English, non-sports related sample $x$. More generally, $w^T [\![\vec{\phi}]\!](x)$ is larger for non-English sports-related samples.

**Defining $p(x \mid \vec{\phi}, w)$.** According to Equation 1, $p(x \mid \vec{\phi}, w)$ is a distribution over $X$, all the text samples we observe, but it puts more weights on $x$ with higher values of $w^T [\![\vec{\phi}]\!](x)$. Using the example $w$ and $\vec{\phi}$ above, $p(x \mid \vec{\phi}, w)$ has higher probability on non-English sports-related texts.

Finally, we define $U(x)$ as the uniform distribution over $X$ for later use.

Figure 2: **Left.** The prompt to compute $[[\phi]](x)$. **Right.** The prompt to `Discretize` $\tilde{\phi}_k$, which generates a set of candidate predicates based on samples $x$ from $U$ and their scores $\cos(e_x, \tilde{\phi}_k)$.

## 3.2 Example Models Parameterized by Natural Language Predicates

We introduce three models parameterized by predicates: clustering, time series, and multi-label classification. For each model, we explain its input, the learned parameters $\vec{\phi}$ and $w$, the log-likelihood loss $\mathcal{L}$, and its relation to classical models.

**Clustering.** This model aims to help humans explore a large corpus by creating clusters, each explained by a predicate. Such a model may help humans obtain a quick overview for a large set of machine learning inputs [60], policy discussions [47], or business reviews [4]. Given a set of text $X$, our model produces a set of $K$ clusters, each parameterized by a learned predicate $\phi_k$; for example, if the predicate is "*discusses the U.S. Election*", then the corresponding cluster is a uniform distribution over all samples in $X$ that discuss the U.S. Election.

Similar to K-means clustering, each sample $x$ is assigned to a unique cluster. We use a one-hot basis vector $b_x \in \mathbb{R}^K$ to indicate the cluster assignment of $x$, and set $w_x = \tau \cdot b_x$, where $\tau$ has a large value (e.g. 10). We maximize the total log-likelihood:

$$\mathcal{L}(\vec{\phi}, w) = -\sum_{x \in X} \log(p(x \mid \vec{\phi}, w_x)); \quad w_x = \tau \cdot b_x, \text{ where } \tau \to \infty \text{ and } b_x \text{ is a basis vector.}$$

However, some samples might not belong to any cluster and thus have 0 probability; to prevent infinite loss, we add another "background cluster" $U(x)$ that is uniform over all samples in $X$; therefore, each sample $x$ can back off to this cluster and incur a loss of at most $-\log U(x) = \log(|X|)$.

**Time Series Modeling.** This model aims to explain latent variations in texts that change across time; for example, finding that an increasing number of people "search about flu symptons" ($\phi$) can help us forecast a potential outbreak [16]. Formally, the input is a sequence of $T$ text samples $X = \{x_t\}_{t=1}^T$. Our model produces $K$ predicates $\phi_k$ that capture the principle axes of variation in $x$ across time. We model $w_1 \ldots w_T$ as being drawn from a Brownian motion, i.e.,

$$p(x_t \mid \vec{\phi}, w_t) \propto \exp(w_t^\top [[\vec{\phi}]](x)); \quad w_t := w_{t-1} + \mathcal{N}(0, \lambda^{-1}I), \tag{4}$$

where $\lambda$ is a real-valued hyper-parameter that regularizes how fast $w$ can change. The loss $\mathcal{L}$ is hence

$$\mathcal{L}(\vec{\phi}, w) = \sum_{t=1}^T -\log(p(x_t \mid \vec{\phi}, w_t)) + \frac{\lambda}{2} \sum_{t=1}^{T-1} ||w_t - w_{t+1}||_2^2. \tag{5}$$

**Multiclass Classification with Learned Feature Predicates.** This model aims to explain the decision boundary between groups of texts, e.g. explaining what features are more correlated with the fake news class [35] compared to other news, or explaining what activates a neuron [5]. Suppose there are $C$ classes in total; the dataset is a set of samples $x_i$ each associated with a class $y_i \in [C]$. Our model is hence a linear logistic regression model on the feature vectors extracted by $\vec{\phi}$, i.e.

$$\text{logits}(x_i) = W \cdot [[\vec{\phi}]](x_i); \quad \mathcal{L}(\vec{\phi}, W) = -\sum_i \log\left(\frac{e^{\text{logits}(x_i)_{y_i}}}{\sum_{c=1}^C e^{\text{logits}(x_i)_c}}\right), \tag{6}$$

where $W \in \mathbb{R}^{C \times K}$ is the weight matrix for logistic regression.

# 4 Method

We can now learn the parameters for each model above by minimizing the loss function $\mathcal{L}$. Formally,

$$\hat{\vec{\phi}}, \hat{w} = \text{argmin}_{\vec{\phi} \in \Phi^K, w} \mathcal{L}(\vec{\phi}, w). \tag{7}$$

However, optimizing $\vec{\phi}$ is challenging, since it is discrete and therefore cannot be directly optimized by gradient-based methods. To address this challenge, we develop a general optimization method, which we describe at a high level in Section 4.1, introduce its individual components in Section 4.2, and explain our full algorithm in Section 4.3.

## 4.1 High-Level Overview

Our framework pieces together three core functions that require minimal model-specific design:

1. OptW, which optimizes $w$.
2. OptRelaxedPhi, which optimizes a continuous relaxation $\tilde{\phi}_k$ for each predicate $\phi_k$.
3. Discretize, which maps from continuous predicate $\tilde{\phi}_k$ to a list of candidate predicates.

Using these three components, our overall method initializes the set of predicates by first optimizing $w$ and $\tilde{\phi}$ using OptW and OptRelaxedPhi and then discretizing $\tilde{\phi}$ with Discretize. To further improve the loss, it then iteratively removes the least useful predicate, re-optimizes its continuous representation, and discretizes it back to a natural language predicate.

To provide more intuition for these three components, we explain what they should achieve in the context of clustering. OptW should optimize the 1-hot choice vectors $w_x$ by assigning each text sample to the cluster with maximum likelihood. OptRelaxedPhi should find a continuous cluster representation $\tilde{\phi}_k$ similar to the sample embeddings assigned to this cluster, and Discretize generates candidate predicates that explain which samples' embeddings are similar to $\tilde{\phi}_k$. Next, we introduce these three components formally for general models with predicate parameters.

## 4.2 Three Components of our framework

OptW optimizes $w$ while fixing the values of $\vec{\phi}$. Formally, $\text{OptW}(\vec{\phi}) := \text{argmin}_w \mathcal{L}(\vec{\phi}, w)$.

This function needs to be designed by the user for every new model, but it is generally straightforward: in the clustering model, it corresponds to finding the cluster that assigns the highest probability for each sample; in classification, it corresponds to learning a logistic regression model; in the time series model, the loss is convex with respect to $w$ and hence can be optimized via gradient descent.

For later use, we define the fitness of a list of predicates $\vec{\phi}$ as the negative loss after $w$ is optimized:

$$\text{Fitness}(\vec{\phi}) := -\mathcal{L}(\vec{\phi}, \text{OptW}(\vec{\phi})). \tag{8}$$

Next, we discuss OptRelaxedPhi. The parameters $\vec{\phi}$ are discrete strings, so the loss function is not differentiable with respect to $\vec{\phi}$. To address this, we approximate $[\![\vec{\phi}]\!](x)$ with the dot product of two continuous vectors, $\tilde{\phi}_k \cdot e_x$, where $e_x \in \mathbb{R}^d$ is a feature embedding of $x$ normalized to unit length (e.g. the last-layer activations of some neural network), and $\tilde{\phi}_k \in \mathbb{R}^d$ is a unit-length, continuous relaxation of $\phi_k$. Intuitively, if the optimal $\phi =$ "*is sports-related*" and $x$ is a sports-related sample with $[\![\phi]\!](x) = 1$, then we hope that $\tilde{\phi}$ would correspond to the latent direction encoding the sports topic and it has high similarity with the embedding $e_x$ of $x$. Under this relaxation, $\mathcal{L}$ becomes differentiable with respect to $\tilde{\phi}_k$ and can be optimized with gradient descent.

Formally, OptRelaxedPhi optimizes all continuous predicates $\tilde{\phi}_{1...K}$ given a fixed value of $w$:

$$\text{OptRelaxedPhi}(w) = \text{argmin}_{\tilde{\phi}_{1:K}} \mathcal{L}(\tilde{\phi} \mid w). \tag{9}$$

We sometimes also use it to optimize a single continuous predicate $\tilde{\phi}_k$ given a fixed $w$ and all discrete predicate variables other than $\phi_k$ (denoted as $\phi_{-k}$):

$$\text{OptRelaxedPhi}(\phi_{-k}, w) = \text{argmin}_{\tilde{\phi}_k} \mathcal{L}(\tilde{\phi}_k \mid \phi_{-k}, w). \tag{10}$$

| Reference | Size | Learned | Size | Surface | F1 |
|-----------|------|---------|------|---------|-----|
| *"artist"* | 0.07 | *"music"* | 0.12 | 0.50 | 0.37 |
| *"animal"* | 0.07 | *"a specific species of plant or animal"* | 0.14 | 0.50 | 0.65 |
| *"book"* | 0.08 | *"literary works"* | 0.07 | 0.50 | 0.64 |
| *"politics"* | 0.06 | *"a political figure"* | 0.06 | 0.50 | 0.96 |
| *"plant"* | 0.07 | *"a specific species of plant or animal"* | 0.14 | 0.50 | 0.68 |
| *"company"* | 0.08 | *"business and industry"* | 0.07 | 0.50 | 0.83 |
| *"school"* | 0.06 | *"schools"* | 0.07 | 1.00 | 0.97 |
| *"athlete"* | 0.07 | *"sports"* | 0.07 | 0.50 | 0.98 |
| *"building"* | 0.08 | *"historical buildings"* | 0.08 | 0.50 | 0.92 |
| *"film"* | 0.06 | *"film"* | 0.07 | 1.00 | 0.91 |
| . . . | . . . | . . . | . . . | . . . | . . . |

Table 1: We compare the reference predicates and our learned predicates when clustering the `DBPedia` dataset. We abbreviate the predicates, e.g. *"art"* = *"has a topic of art"*. For each reference, we match it with the learned predicate that achieves the highest F1-score at predicting the reference denotation. We also report the surface similarity (defined in Section 5.2) between the learned predicate and the reference. Our learning algorithm mostly recovers the underlying reference predicates, though it sometimes learns larger/correlated cluster that disagrees with the reference but is still meaningful.

Finally, `Discretize` converts $\tilde{\phi}_k$ into a list of $M$ discrete candidate predicates to update the variable $\phi_k$. Our goal is to find $\phi$ whose denotation is highly correlated with the dot product simulation $\tilde{\phi}_k \cdot e_x$.

To discretize $\tilde{\phi}_k$, we prompt a language model to generate several candidate predicates and then re-rank them. Concretely, we draw samples $x \sim U(x)^3$ and sort them based on their dot product $\tilde{\phi}_k \cdot e_x$. We then prompt a language model with these sorted samples and ask it to generate candidate predicates that can explain what types of samples are more likely to appear later in the sorted list (Figure 2 bottom). To filter out unpromising predicates, we re-rank them based on the pearson-r correlations between $[\![\phi]\!]$ and $\tilde{\phi}_k \cdot e_x$ on $U$ if $w$ cannot be negative (e.g. clustering), and the absolute value of pearson-r correlation otherwise. We then keep the top-$M$ predicates.

### 4.3 Piecing the Three Components Together

Our algorithm has two stages: we first initialize all the predicate variables and then iteratively refine each of them. During initialization, we

1. randomly initialize continuous predicates $\tilde{\phi}$ to be the embedding of random samples from $X$
2. optimize $\mathcal{L}(\tilde{\phi}, w)$ by alternatively optimizing $w$ and all the continuous predicates $\tilde{\phi}$ with `OptW` and `OptRelaxedPhi`, and
3. set $\phi_k$ as the first candidate from `Discretize`$(\tilde{\phi}_k)$

During refinement, we repeat the following steps for $S$ iterations:

1. find the least useful predicate $\phi_k$; we define the usefulness of $\phi_k$ as how much the fitness would decrease if we zero it out, i.e. $-\texttt{Fitness}(\vec{\phi}_{-k}, 0)$.
2. optimize $\tilde{\phi}_k$ using `OptRelaxedPhi` and choose the fittest predicate from `Discretize`$(\tilde{\phi}_k)$

We include a formal description of our algorithm in Appendix Algorithm H.

## 5 Experiments

In this section, we benchmark our algorithm from Section 4; we later apply it to open-ended applications in Section 6. We run our algorithm on datasets where we know the ground truth predicates $\vec{\phi}$ and evaluate whether it can recover them. On five datasets and three statistical models, continuous relaxation and iterative refinement consistently improve performance. Our general method also matches a previous specialized method for explainable clustering [53].

---

[3] i.e. uniformly draw samples $x$ from all samples we observe from the dataset

## 5.1 Datasets

We design a suite of datasets for each of the three statistical models mentioned from Section 3.2. Each dataset has a set of reference predicates, and we evaluate our algorithm's ability to recover them.

**Clustering.** We consider five datasets, `AGNews`, `DBPedia`, `NYT`, `Bills`, and `Wiki` [40, 58, 32, 23]. The datasets have 4/14/9/21/15 topic classes each described in a predicate, and we sample 2048 examples from each for evaluation.

**Multiclass Classification.** We design a classification dataset with 5,000 articles and 20 classes; its goal is to evaluate a method's ability to recover the latent interpretable features useful for classification. Therefore, we design each class to be a set of articles that satisfy three predicates about its topic, location, and language; for example, one of the classes can be described by the predicates "*has a topic of sports*", "*is in Japan*", and "*is written in English*". We create this dataset by adapting the New York Times Articles dataset [40], where each article is associated with a topic and a location predicate; we then translate them into Spanish, French, and Deutsch. We consider in total $4 + 4 + 4 = 12$ different predicates for each of the topic/location/language attributes and subsample 20 classes from all $4 \times 4 \times 4 = 64$ combinations.

**Time Series modeling.** We synthesize a time series problem by further adapting the translated NYT dataset above. We set the total time $T = 2048$ and sample $x_1 \ldots x_T$ according to the time series model in Section 3.2 to create the benchmark. We set $\vec{\phi}$ to be the 12 predicates mentioned above and the weight $w_{\cdot,k}$ for each predicate $\phi_k$ to be a cosine function with a period of $T$ to simulate how each attribute evolves throughout time. In addition, we included three simpler datasets where there is only variation on one attribute (i.e. varies only on one of topic/location/language). We name these four time series modeling `all`, `topic`, `locat`, and `lang`, respectively. See Appendix B for a more detailed explanation.

## 5.2 Metrics

To evaluate our algorithm, we match each learned predicate $\hat{\phi}_k$ with a reference $\phi^*_{k'}$, compute the F1-score and surface similarity for each pair, and then report the average across all pairs. To create the matching, we match $\hat{\phi}_k$ to the $\phi^*_{k'}$ with the highest overlap (number of samples where both are true); formally, we define a bi-partite matching problem to match each predicate in $\hat{\phi}$ with one in $\phi^*$, define the weight of matching $\phi^*_{k'}$ and $\phi^*_{k'}$ to be their overlap, and then find the maximum weight matching via the Hungarian algorithm. We now explain the F1-score and surface similarity metric.

**F1-score Similarity.** We compute the F1-score of using $\hat{\phi}(x)$ to predict $\phi^*(x)$ on $X$, the set of samples we observe. This is similar to the standard protocol for evaluating cluster quality [28].

**Surface Form Similiarity**. We can also directly evaluate the similarity between two predicates based on their string values, e.g. "*is about sports*" is similar in meaning to "*has a topic of sports*", a metric previously used by [62]. For a pair of predicates, we ask `gpt-4` to evaluate whether they are similar in meaning, related, or irrelevant, with each option associated with a surface-similarity score of 1/0.5/0. We display the prompt in Figure 5 and example ratings in Table 1.

## 5.3 Experiments on Our Benchmark

We now use these metrics and datasets to evaluate the optimization algorithm proposed in Section 4 and run ablations to investigate whether continuous relaxation and iterative refinement are effective. We will first introduce the overall experimental setup, and then discuss individual takeaways supported by experimental results in each paragraph.

**Experimental Setup.** When running the algorithm, we generate candidate predicates in `Discretize` with `gpt-3.5-turbo` [37]; to perform the denotation operation $[\![\phi]\!](x)$, we use `flan-t5-xl` [13]; we create the embedding for each sample $x$ with the `Instructor-xl` model [48] and then normalize it with $\ell_2$ norm. We set the number of candidates $M$ returned by `Discretize` to be 5 and the number of optimization iteration $S$ to be 10. To reduce noises due to randomness, we average the performance of five random seeds for each experiment.

Table 2 reports the results of clustering and Table 3 reports other results. For each dataset, we perform several ablation experiments and present the takeaways from these results.

| F1/Surface | AGNews | DBPedia | NYT | Bills | Wiki | Average |
|---|---|---|---|---|---|---|
| Prompting | 0.43/0.60 | 0.31/0.44 | 0.21/0.40 | 0.16/0.47 | 0.22/0.34 | 0.27/0.45 |
| No-Refine | 0.72/0.57 | 0.57/0.52 | 0.54/0.58 | 0.34/0.49 | 0.47/0.51 | 0.53/0.54 |
| No-Relax | 0.86/0.60 | 0.59/0.53 | 0.58/0.53 | 0.31/0.51 | 0.46/0.50 | 0.56/0.54 |
| Ours | **0.86**/0.62 | 0.68/0.54 | **0.70/0.63** | **0.45/0.52** | **0.51/0.53** | **0.64/0.57** |
| GoalEx (Specialized) | **0.86/0.62** | **0.75/0.64** | 0.68/**0.63** | 0.33/0.50 | 0.49/0.48 | 0.62/**0.57** |

Table 2: Results on clustering. `Ours` always outperforms `No-Refine` and `No-Relax`, indicating that both continuous relaxation and iterative refinement are helpful. Compared to `GoalEx` [53], our method is slightly better on all datasets except `DBPedia`, which we analyze in Table 1.

| F1/Surface | topic | lang | locat | all | time-avg | classification |
|---|---|---|---|---|---|---|
| Prompting | 0.40/0.35 | 0.39/0.38 | 0.26/0.30 | 0.54/0.57 | 0.40/0.40 | 0.51/0.42 |
| No-Refine | 0.53/0.53 | 0.39/0.50 | 0.37/0.55 | 0.58/0.44 | 0.47/0.50 | 0.58/0.44 |
| No-Relax | 0.65/0.50 | 0.52/0.65 | 0.48/**0.68** | 0.61/0.56 | 0.56/0.60 | 0.68/0.62 |
| Shuffled | 0.46/0.33 | 0.52/0.45 | 0.33/0.28 | 0.60/0.39 | 0.47/0.35 | N/A |
| Ours | **0.67/0.57** | **0.62/0.70** | **0.55/0.68** | **0.72/0.64** | **0.64/0.65** | **0.73/0.70** |

Table 3: Our performance on time series (left) and classification (right). Both continuous relaxation and iterative refinement improve the performance (comparing `Ours` to `No-Refine` and `No-Relax`).

**Takeaway 0: Is our method better than naïvely prompting language model to generate predicates?** How does our approach compare to a naïve baseline approach, which directly prompts the language model to generate predicates based on dataset samples? For this baseline, we repeatedly prompt a language model to generate more predicates until we obtain $K$ predicates, compute their denotation, evaluate them using the metrics in Section 5.2, and report the performance in Table 2 and 5, the `Prompting` row. Across all entries, our approach significantly outperforms this baseline.

**Takeaway 1: Relax + discretize is better than exploring randomly generated predicates.** Our optimization algorithm explores the top-5 LLM-generated predicates that have the highest correlations with $\tilde{\phi}_k \cdot e_x$. Would choosing a random predicate be equally effective? To investigate this question, we experimented with a variant of our algorithm that randomly chooses five predicates without utilizing the continuous representation $\tilde{\phi}_k$ (No-Relax). In Table 2 and 3, `No-Relax` underperforms our full algorithm (`Ours`) in all cases. In Appendix Figure 6, we plot the loss after each iteration averaged across all tasks, and we find that `Ours` converges much faster than `No-Relax`.

**Takeaway 2: Iterative refinement improves the performance.** We considered a variant of our algorithm that only discretizes the initial continuous representations and does not iteratively refine the predicates (No-Refine). In Table 2 and 3, `No-Refine` underperforms the full algorithm in all cases.

**Takeaway 3: Our model-agnostic method is competitive with previous methods specialized for explainable clustering.** We compare our method to `GoalEx` from [53], which designs a specialized method for explainable clustering based on integer linear programming. Even though our method is model-agnostic, it matches or outperforms `GoalEx` on four out of five datasets and improves F1 by 0.02 on average.

**Takeaway 4: Our method accounts for information beyond the set of text samples (e.g. temporal correlations in the time series).** We investigate this claim using the time series datasets, where we shuffle the text order and hence destroy the time-dependent information a model could use to extract informative predicates (Shuffled). Table 3 finds that `Ours` is better than `Shuffled` in all cases, indicating that our method does make use of temporal correlations.

Appendix D includes additional results: 1) compared to topic modeling and K-means, our method achieves similar or better performance while being explainable; 2) we ran ablations on the effect of neural embeddings and show that informative embeddings are crucial to good performance; 3) Takeaways 1, 2, and 4, are significant with $p < 1\%$ under paired t-tests.

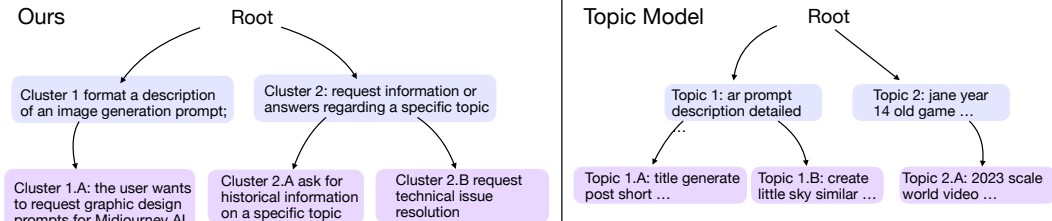

Figure 3: **Left.** We generate a taxonomy with sophisticated explanations by recursively applying our clustering model. **Right.** We cluster with topic models and present the top words for each topic. Although some topics are plausibly related to certain applications, they are still ambiguous.

## 6 Open-Ended Applications

We apply our framework to a broad range of applications to show that it is highly versatile. Our framework can monitor data streams (Section 6.1), apply to the visual domain (Section F.1), and be easily steered to explain specific abstract properties (Section F.2). Across all applications, our framework is able to explain complex concepts that classical methods struggle to produce.

### 6.1 Running Our Models Out of the Box: Monitoring Complex Data Streams of LLM Usages

We apply our models from Section 3.2 to monitor complex data streams of LLM usages. In particular, we recursively apply our clustering model to taxonomize user queries into application categories, apply our time series model to characterize trends in use cases across time, and apply our classification model to find categories where one LLM is better than the other. Due to space constraints, we present the key results in the main paper and the full results in Appendix G.

**Taxonomizing User Applications via Clustering.** LLMs are general-purpose systems, and users might applyLLMs in ways unanticipated by the developers. If the developers can better understand how theLLMs are used, they could collect training data correspondingly, ban unforeseen harmful applications, or develop application-specific methods. However, the amount of user queries is too large for individual developers to process, so an automatically constructed taxonomy could be helpful.

We recursively apply our clustering model to user queries to the ChatGPT language model. We obtain the queries by extracting the first turns from the dialogues in WildChat [59], a corpus of 1M real-world user-ChatGPT dialogues. We use `gpt-4o` [36] to discretize and `claude-3.5-sonnet` [1] to compute denotations. We first generate $K = 6$ clusters on a subset of 2048 queries; then we generate $K = 4$ subclusters for each cluster with $> 32$ samples.

We present part of the taxonomy in Figure 3 (left) and contrast it with the taxonomy constructed by directly applying LDA recursively (right). Although some LDA topics are plausibly related to certain applications, they are still ambiguous; for example, it is unclear what topic 1 "*ar prompt description detailed*" means. After manually inspecting the samples associated with this topic, we found that they were related to the application of writing prompts for an image-generation model. In contrast, our framework can explain complicated concepts that are difficult to infer from individual words; for example, it generates "*requesting graphic design prompts*" for the above application, which is much clearer in its meaning when explained in natural language.

**Characterizing Temporal Trends via Time Series Modeling.** Understanding temporal trends in user queries can help forecast flu outbreaks [16], prevent self-reinforcing trends [19], or identify new application opportunities. We run our time series model on 1000 queries from WildChat with $K = 4$ to identify temporal trends in user applications, and report part of the results in Figure 4. Based on the blue curve, we find that an increasing number of users "*requests writing or content creation .... creating stories based on given prompts.*'. This helps motivate systems like Coauthor [29] to assist with this use case.

**Finding Categories where One Language Model is Better than the Other.** One popular method to evaluateLLMs is crowd-sourcing: an evaluation platform (e.g. ChatBotArena [11]) / or a company (e.g. OpenAI) accepts prompts from users, shows users responses from two different LLM systems,

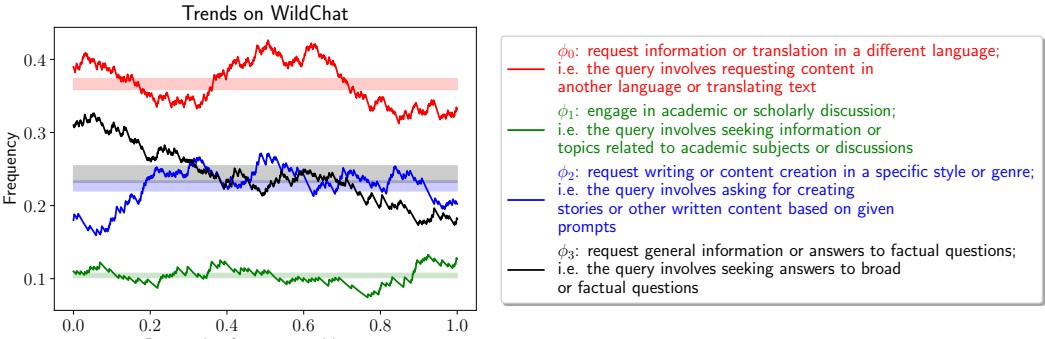

Figure 4: We analyze WildChat queries with our time series model. For each learned predicate, we plot how its frequency evolves and the 99% confidence interval of the average frequency (shaded).

and the users indicate which one they like better. The ranking among the LLM systems is then determined by Elo-rating, i.e. how often they win against each other.

However, aggregate Elo-rating omits subtle differences between LLM systems. For example, LLama-3-70B achieved a similar rating as Claude-3-Opus, and the LLM community was excited that open-weight models were catching up. However, is LLama-3-70B similarly capable across all categories, or is it significantly better/worse under some categories? Such information is important for downstream developers, since some capabilities are more commercially valuable than others: e.g. a programmer usually does not care about LLM's capability to write jokes. We need a more fine-grained comparison.

We directly apply the classification model from our framework to solve this task. To understand the categories where LLama-3-70B is better/worse than Claude-3-Opus, we gather user queries $x$ from the ChatBotArena maintainers (personal communication), set $y = 1$ if the LLama-3-70B's response to $x$ is preferred and $y = 0$ otherwise. We set $K = 3$.

Our model finds that LLama-3-70B is better when the query "*asks an open-ended or thought-provoking question*" but worse when it "*presents a technical question*" or "*contains code snippets*". These findings are corroborated by manual analysis by the ChatBotArena maintainers, who also found that Llama-3 is better at open-ended and creative tasks while worse at technical problems[4]. We hope that our model can automatically generate similar analysis in the future when a new LLM is released, thus saving researchers' efforts.

To summarize, our framework 1) enables us to define a time series model to explain temporal trends in natural language, and 2) outputs sophisticated explanations that LDA fails to generate. However, it is far from perfect: it is slow to compute denotations for all pairs of $x$ and candidates $\phi$ since it involves many LLM API calls, and the predicates themselves are sometimes redundant. We describe these limitations and potential ways to improve them in Appendix G.

Due to space constraints, we present applications in explaining visual features to make images memorable to humans and clustering math problems based on subareas in Appendix F.1 and F.2.

# 7 Conclusion

In this work, we formalize a broad family of models parameterized by natural language predicates. We design a learning algorithm based on continuous relaxation and iterative refinement, both of them effective based on our ablation studies. Finally, we apply our framework to a wide range of applications, showing that it is highly versatile, practically useful, applicable to both text and vision domains, and explains sophisticated concepts that classical methods struggle to produce. We hope future works can make our method more computationally efficient and apply it to more realistic applications, thus assisting humans to discover and understand complex patterns in the world.

---

[4]`https://lmsys.org/blog/2024-05-08-llama3/`

## Acknowledgement

Ruiqi Zhong designed the conceptual framework, the algorithm, and all the experiments; he also implemented all the experiments and wrote the entire paper. Heng Wang contributed to earlier explorations of the project. Dan Klein provided feedback on the paper draft. Jacob Steinhardt provided feedback throughout the project.

Ruiqi Zhong is supported by Simons Foundation fund, chartstring 71815-13090-44–PSJST. We thank members from the Berkeley NLP group, Jacob Steinhardt group, Lisa Dunlap, Zihan Wang, Tatsunori Hashimoto, and anonymous reviewers for their feedback on our project and paper draft.

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

# A   More Related Work

**LLM for Exploratory Analysis.** Due to its code generation capability [10], large language models have been used to automatically generate programs to analyze a dataset and generate reports from them [31, 20]. In comparison, our work focuses on generating natural language parameters to extract real-valued features from structured data.

**Discrete Prompt Optimization.** Many prior works optimized discrete prompts to improve the predictive performance [43, 14], and some recent works demonstrated that LLMs can optimize prompts to reach state-of-the-art accuracy [64, 56]. In comparison, we focus on optimizing discrete predicates to explain patterns rather than improve task performance.

**Learning with Latent Language.** [3] first proposed to learn in a hypothesis space of natural language strings to improve generalization, and later works in this area have focused on using natural language to guide the learning process to improve downstream task performance [33, 25, 42, 54]. In contrast, our work focuses on explaining datasets with natural language, rather than improving downstream task performance.

# B   Time Series Dataset

To sample texts from the `All` time series problem, we sample from the time series model described in Section 3.2; we set $\vec{\phi}$ to be all the 12 predicates, sort them first by attributes (e.g. topic/location/language) then alphabets, and we set the weight for the $k^{\text{th}}$ predicate to be a sin function with period $T$ and evenly spaced phases, i.e.

$$w_{k,t} = \sin(2\pi(\frac{t}{T} + \frac{k}{K}))  \tag{11}$$

As a result, the weight for each predicate has evenly spaced phases and would peak at different time period.

# C   Surface form similarity prompt

We include our prompt used to evaluate the surface form similarity between the predicted predicate $\hat{\phi}_k$ and the reference predicate $\phi_k^*$ in Figure 5.

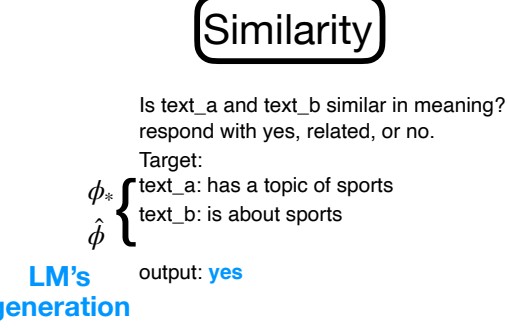

Figure 5: The prompt template used to evaluate the surface form similarity between the predicted predicate $\hat{\phi}_k$ and the reference predicate $\phi_k^*$.

# D   Additional Results on Our Benchmark

**Our method is similar or better than classical methods such as topic modeling or K-means.** We report the performance of K-means clustering and topic modeling under the clustering benchmark

| F1/Surface | AGNews | DBPedia | NYT | Bills | Wiki | Average |
|---|---|---|---|---|---|---|
| OneHot | 0.87/0.53 | 0.54/0.51 | 0.48/0.53 | 0.26/0.51 | 0.36/0.47 | 0.50/0.51 |
| OtherEmb | 0.85/0.70 | 0.62/0.54 | 0.59/0.53 | 0.43/0.59 | 0.48/0.53 | 0.60/0.59 |
| Ours | 0.86/0.62 | 0.68/0.54 | 0.70/0.63 | 0.45/0.52 | 0.51/0.53 | 0.64/0.57 |
| K-means | 0.83/—– | 0.75/—- | 0.72/—- | 0.41/—- | 0.53/—- | 0.65/—- |
| TopicModel | 0.56/—– | 0.52/—- | 0.49/—- | 0.25/—- | 0.35/—- | 0.43/—- |

Table 4: We compare our method to classical clustering approaches that do not generate natural language explanations (`K-means` and `TopicModel`), where "—–" means that the surface form metric is undefined since these methods do not output natural language explanations. We find that on average, our method is close to `K-means` and significantly outperforms `TopicModel` under the F1 similarity metric, while generating natural language explanations for each cluster. We also compare our method to using one-hot text embedding, and find that our method is significantly better; this indicates that the use of informative text embedding is crucial to performance.

| F1/Surface | topic | lang | locat | all | time-avg | classification |
|---|---|---|---|---|---|---|
| One-hot | 0.63/0.55 | 0.51/0.57 | 0.51/0.62 | 0.66/0.60 | 0.58/0.59 | 0.72/0.68 |
| OtherEmb | 0.68/0.58 | 0.56/0.59 | 0.49/0.68 | 0.71/0.68 | 0.61/0.63 | 0.73/0.67 |
| Ours | 0.67/0.57 | 0.62/0.70 | 0.55/0.68 | 0.72/0.64 | 0.64/0.65 | 0.73/0.70 |

Table 5: Our method consistently outperforms a variant that uses one-hot text encoding as $e_x$ rather than neural embeddings. This indicates that using informative text embedding is crucial to performance.

in Table 4. on average, our method is close to `K-means` and significantly outperforms `TopicModel` under the F1 similarity metric, while generating natural language explanations for each cluster.

**Takeaway 5: Using informative text embedding is crucial to performance.** We used neural embeddings when optimizing the continuous representation of the predicates. Does our algorithm actually make use of the information in the feature embeddings? To investigate this question, we ran an ablation of using one-hot text embeddings rather than neural embeddings (`OneHot`), which do not encode any information about the similarity between text samples. We report the performance in Table 4 and 5; across all settings, using neural embeddings consistently outperforms `OneHot`.

To make sure that this takeaway is general and not specific to one embedding model, we run our method with another text embedding model, `all-mpnet-base-v2`[5] and report the performance as

---

[5] https://huggingface.co/blog/1b-sentence-embeddings

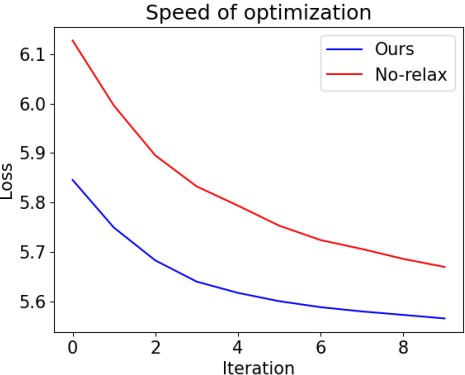

Figure 6: We plot how the loss decreases across different iterations with and without relaxation (that explores using random predicates). We find that using relaxation significantly speeds up optimization.

the `OtherEmb` row. We find that using this neural embedding also outperforms `OneHot` in most cases, indicating that our conclusion is robust.

**Takeaway 1,2,and 4** are statistically significant. To compare the performance between our method and each variant, we conduct a one-sided paired t-test on their performance (F1-similarity) on each dataset, where the performance on each dataset is the averaged performance across five runs. Takeaway 1, 2, 4 has a $p$-value of $5 \times 10^{-4}$, $2 \times 10^{-4}$, and $6 \times 10^{-3}$, respectively.

# E    Detecting Self-Reinforcing Trends in Machine Learning System

Machine learning systems sometimes have unintended side effects and reinforcement themselves. [19] illustrated an example failure mode, where a group of users is discriminated against and thus leave a platform, causing a ML system to discriminate them further and hence drive them away.

As a concrete illustration, let us imagine a social platform Y where users post tweets and the platform will display the most engaging ones; suppose there are two groups of users, one conservative and one liberal, where both groups prefer more engaging tweets but also tweets that agree with their political stances. Y implements a recommender system, which trains a classifier to predict whether a tweet is likely to be preferred by a random user, and then the platform Y will promote these tweets. If the two groups of users are balanced, the optimal classifier will make Y promote tweets that are engaging and place little weights on the political slant.

However, if there are fewer liberal users, the classifier will be biased and Y will promote conservative tweets more and focus less on whether the tweet is engaging or not. The liberal users will find the promoted tweets less attractive, thus leaving the platform Y. As a result, fewer liberal users will stick to Y, thus making the classifier more biased.

Now we provide a proof-of-concept experiment to illustrate how our time series model can be applied to detect such a reinforcing trend. We first simulate the setup above and obtain the tweets promoted by platform Y across time, and then apply our time series model to extract temporal trends from these tweets. Suppose there are two groups of users, liberal and conservative. At $t = 0$, the fraction of liberal users is $\lambda_0 = 0.5$ and is the same as that of conservative users. To simulate the setup above and obtain the tweets promoted by platform Y across time, we assume that at each time step $t$, we will sample 2,000 tweets, where each tweet is a 2D datapoint with the $x$-value a random integer from [-1, 1] indicating whether it is liberal, non-political, or conservative, and $y$-value a random integer from [-2, 2] indicating how engaging the tweet is. For each tweet, we obtain a label of $y = 1/0$ if the user likes a tweet, and the user's probability for liking a tweet is defined by $\sigma(ux + 0.5y)$, where $u = 1$ if the user is conservative and 0 otherwise. We then train a logistic regression classifier to predict whether a random user will like a tweet and the platform $Y$ will promote the tweets with the top 20% score. Let the fraction of tweets non-liberal tweets be $a_t$ and non-conservative tweets be $b_t$, then the fraction of liberal users for the next round will be determined by:

$$\lambda_{t+1} = \frac{b_t \lambda_t}{b_t \lambda_t + a_t(1 - \lambda_t)}, \tag{12}$$

which models how the group size will increase/decrease depending on whether the platform promotes tweets that agree with their views. We run this process for $T = 10$ and gather all the 2D datapoints promoted by platform Y.

We then turn these two-dimensional datapoints into text samples $x$. We ask the gpt-4o to write a liberal, non-political, or conservative tweet based on the $x$-value; then we ask gpt-4o to make it more/less engaging based on the $y$-value. For example, for a 2D value of (-1, 2), we ask gpt-4o to write a liberal tweet and ask it to make it more engaging two times; if the value is (1, -2), we ask gpt-4o to write a conservative tweet and then ask it to make it less engaging two times.

We now have a list of tweets across time, and we directly apply our time series model with $K = 3$ to extract trends from them. Our time series model find that there is an increasing amount of tweets that "*expresses patriotic sentiments*" and "*champions specific policies*", but a decreasing amount "*poses a question to engage the audience*". These predicates exactly recover all the underlying trends, that the self-reinforcing effect make the tweets more conservative, less non-political, and less engaging.

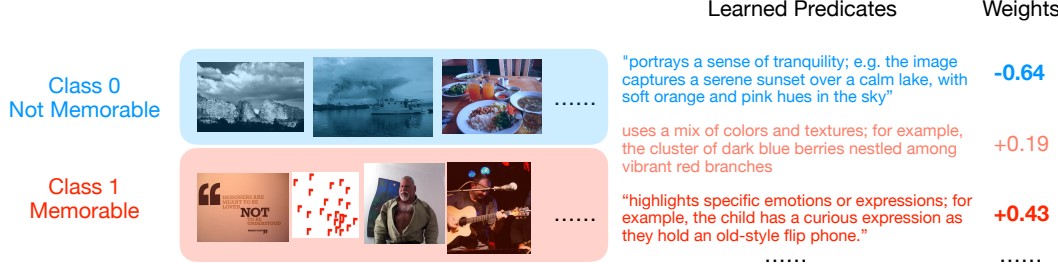

Figure 7: We apply our classification model from Section 3.2 to explain what visual features make images more memorable [24]. Consistent with previous findings, we find that tranquil scenes make an image less memorable, while emotions and expressions are more memorable.

| Ours | Classical Method |
|---|---|
| 1. involves algebraic manipulation; | 1. asy, draw, axis, operatorname, tabular → Unclear what this means |
| 2. involves probability or combinatorics | 2. divisors, probability, many, letters, unique → Maybe combinatorics |
| 3. requires geometric reasoning; | 3. decimal, det, compute, evaluate, power → Probably algebra |
| 4. pertains to number theory; | 4. solutions, roots, polynomial, solution, minimum → Another algebra cluster? |
| 5. involves calculus or limits; | 5. hyperbola, corresponds, proportional, vertices, points → Maybe geometry |
| ✔ Directly explainable | ? Vague |

Figure 8: We cluster the MATH dataset [21] and compare our method (left) to a classical method (right), which first clusters via K-means and then explains each cluster via unigram analysis. Our method directly explains complex concepts, while the classical method delivers vague explanations.

# F   More Applications

## F.1   Applying Our Classification Model to Images: Explaining Memorable Visual Features

Our framework is applicable to the vision domain since a natural language predicate $\phi$ can extract binary values from an image $x$. For example, for the rightmost image $x$ in Figure 7 right, the predicate "*portrays a person*" evaluates to 1, i.e. $[\![\phi]\!](x) = 1$, while "*contains texts*" evaluates to 0.

We present an application of our classification model from Section 3.2 to images, which learns linear weights over a set of visual features described by natural language predicates. This model has also appeared in prior works: our model is equivalent to the language-based concept bottleneck model proposed by [55, 41]; additionally, when $K = 1$ and $C = 2$, our model is equivalent to the VisDiff framework [15], which finds a single predicate to discriminate samples from two classes of images.

We apply our classification model to the LaMem dataset [24] to understand what visual features make an image more memorable, an interesting cognitive science question. We now define the samples $x_i$ and their class labels $y_i$ to run our classification model. In LaMem, each image is associated with a score of how memorable it is as measured by whether humans can remember seeing it in the past; to make implementation easier, we set $x_i$ to be the caption of the image and $y_i = 1$ if $x_i$ has an above median score and $y_i = 0$ otherwise. To fit our classification model, we set $K = 6$, use `gpt-4o` as the discretizer, and use `gpt-4o-mini` to compute denotation.

We present three learned predicates in Figure 7. We find that an image is less memorable if it "*portrays a sense of tranquility; e.g. the image captures a serene sunset over a calm lake, with soft orange and pink hues in the sky ...*", and more likely to be memorable if it "*highlights specific emotions or expressions; for example, the child has a curious expression ...*". These results are consistent with the previous manual analysis from [24], suggesting the validity of our results.

## F.2   Explaining Abstract Properties via Easy Steering: Clustering Problems Based on Subarea

Can our framework explain more abstract aspects of a sample $x$: e.g. subarea, the type of knowledge required to solve a math problem $x$? We show this is feasible by applying our model from Section 3.2 to cluster math problems and steering it to focus on explaining subareas. Meanwhile, classical methods struggle to explain abstract aspects.

We apply our clustering model from Section 3.2 to cluster the MATH dataset [21] based on subareas. The MATH dataset contains five labeled subareas[6] and we hope our model can recover all of them: `Algebra`, `counting_and_probability`, `geometry`, `number_theory`, and `precalculus`. To steer our clustering model to explain subareas, we simply prompt the discretizer LLM "*I want to cluster these math problems based on the type of skills required to solve them.*" We set $K = 5$, using `gpt-4o` to discretize and `gpt-4o-mini` to compute denotation.

We present the outputs of our model on the left of Figure 8. With simple prompting, our model is successfully steered to cluster based on subareas and recovers all five labeled subareas from the MATH dataset. Note that our explanations can explain abstract properties that have no word overlap with the samples that match them: for example, the math problems that "*requires geometric reasoning*" (Figure 6 left 3) usually contain neither of the word "geometric" or "reasoning".

We compare our method to a classical baseline that first clusters the samples and then explains each cluster with representative words. In this baseline, we first perform K-means clustering on the neural embeddings of $x$ and assign each sample to a cluster; we then extract representative words by first running a unigram regression to predict whether a sample belongs to the cluster and then selecting words with the most positive weights. We present the word-based explanations on the right in Figure 8. Overall, significant guesswork is needed to interpret the meaning of each word-based cluster (e.g., it is unclear what cluster 1 represents in Figure 8 right), while the predicates generated by our algorithm are directly explainable. Our framework can be steered to explain more abstract aspects of a sample $x$ and significantly improve over classical methods.

## G   Implementation Details on Open-Ended Applications

To obtain the best outputs, we discretize with `gpt-4o` and compute denotations with `claude-3.5-sonnet`. Since we aim to analyze user queries, we explicitly prompt `gpt-4o` to generate detailed predicates about use cases when discretizing continuous predicates. See 9 for the full prompt.

### G.1   Taxonomizing User Applications.

**Implementation Details** We cluster 1024 dialogue with $K = 6$ and $S = 5$ We only cluster on a small set of dialogue turns because it is slow to compute denotations: 1) we in total explored around 100 predicates and this amounts to $\sim 100 \times 1024 = 100$K LLM API calls, and 2) we used language model API (`claude-3.5-sonnet`), rather than a local small language model (`google/flan-t5-xl`), to compute denotations, since this is the cheapest model that we feel confident that it can handle more sophisticated predicates.

**Full Results.** We present the full results in Figure 10 Overall, we find that our framework can generate sophisticated explanations that classical methods cannot generate. However, some cluster descriptions are significantly overlapping (e.g., category 0, 1, and 0.D); additionally, some sub-clusters are not indeed subsets of their parent categories (e.g., subcategory 2.D does not belong to category 2). Future work can improve the taxonomy by 1) deduplicating semantically similar descriptions or more heavily penalizing cluster overlaps, and 2) steer the predicate generation process so that the descriptions for the subclusters are indeed subsets of their parent descriptions.

### G.2   Characterizing Temporal Trends.

**Implementation Detail.** We run our time series model on 1K dialogue turns with $K = 4$ and the number of iterations $S$ to be 10 to identify temporal trends in user applications. We obtain the smoothed frequency curve by updating with the follow equation:

$$f_t = 0.99 \cdot f_{t-1} + 0.01 [\![\phi_k]\!](x_t); \quad f_0 = \frac{1}{100} \sum_{t=1}^{100} [\![\phi_k]\!](x_t) \tag{13}$$

We obtain the shaded area in Figure 4 by shuffling $x_t$ and find the highest and lowest f values across 100 random runs.

---

[6]after merging similar categories that differ in levels of difficulty

```
Here is a corpus of user queries each associated with a score. The queries are sorted from the lowest to
 the highest score.

{samples_in_prompt_w_scores}

I am a machine learning researcher that builds chat bots. Here is a list of first turns of user queries,
 and I want to cluster them based on their applications. Note that I am only interested in applications:
 for example 'refers to pop culture' is not an application, but 'wants to ask for information about a po
p culture entity' is an application. Each description should start with 'the user wants to ....'

We want to understand what kind of queries achieve a higher score, so please suggest descriptions about
the queries that are more likely to achieve higher scores.
Please suggest me at most {num_descriptions_per_prompt} descriptions, one in a line, starting with "-" a
nd surrounded by quotes "". Each of them needs to be a predicate about a query followed by an explanatio
n and an example that satisfies the predicate, for example:
- "the user wants to request for email or message composition; specifically, the query involves a reques
t for drafting an email or message with specific content and intent. For example, 'write an email to ter
minix llc with a proposal for cooperation...'."
- "the user wants to request technical code/script writing; specifically, the query demands the creation
 or modification of a script or code. For example, 'create a dockerfile based on this script.'"

Do not output anything else. Please do not mention score in your example. (Note that the examples might
not be goal related, and your response should be both formatted correct as above and related to the goal
.)

Please generate the response based on the given datapoints as much as possible. We want the descriptions
 to be relatively objective and can be validated easily, e.g. "is surprising" means different things for
 different people, so we want to avoid such descriptions. It should also be a predicate on a single quer
y (rather than a statement about a comparison); for example, instead of saying "uses more polite languag
e", the generation should be "uses polite language". Sometimes KeyInfo is provided to help you make come
 up with better responses (though it might also be unavailable).

Again, here's the goal.
I am a machine learning researcher that builds chat bots. Here is a list of first turns of user queries,
 and I want to cluster them based on their applications. Note that I am only interested in applications:
 for example 'refers to pop culture' is not an application, but 'wants to ask for information about a po
p culture entity' is an application. Each description should start with 'the user wants to ....'

Your responses are:
- "
```

Figure 9: A discretizer prompt that explicitly asks LLM to explain user applications. E.g., at the end
of the prompt, we explicitly requested the predicates to start with "*the user wants to...*".

## G.3   Advantages and Limitations

Overall, we find that our framework allows us to define sophisticated models (e.g. time series) and
can output highly sophisticated predicates, which can include detailed explanations and examples.
Therefore, when implemented perfectly, its utility has a much higher upperbound than classical
methods such as n-gram Bayes/regression or topic models.

However, the comparison between our method and classical methods is only qualitative: we only eye-
balled the outputs from our method and the classical methods in Section 6 and did not quantitatively
measure how useful they are in practice. Therefore, even if our method does outperform classical
methods such as topic model on our benchmark (Table 4), it might not directly translate to how useful
it is in real-world applications. Additionally, we did not compare to modern taxonomy construction
method such as [57], which involves a lot of task-specific engineering; our method is model-agnostic
and was applied out-of-the-box to construct the taxonomy. Section 6 only shows that our method can
generate more sophisticated natural language explanations, which presents a higher upperbound of
what our method could potentially achieve.

In terms of the weakness of our method, our method is currently slow, as its performance highly
depends on LLMs to compute denotations correctly, it outputs semantically similar predicates that
add little information, and it is hard to control the predicates to satisfy certain properties (e.g. being a
subset of a parent category). We look forward to future works that can address these problems and
realize the full potential of this framework. For example, to remove similar predicates, one could
prompt a language model to check the pairwise surface similarity between two predicates; to speed
up inference, one can distill a smaller but much more efficient model specialized for computing
denotations.

```
0 the user wants to request a descriptive screenplay writing; specifically, the query asks for the generation of screenplays including dialogues and deta
iled background. For example, 'Write a very long, coherent, elaborate, descriptive and detailed screenplay...'
    0.A the user wants to request narrative writing from a personal or first-person perspective; specifically, the query asks for detailed descripti
ons and thoughts from the viewpoint of a character. For example, 'Write a long, detailed, original, imaginative and interesting scene narrated by Celesti
ne from the first person perspective.'
    0.B the user wants to request role-playing; specifically, the query involves engaging in a narrative role-playing scenario with predefined chara
cters. For example, 'We'll role play as characters from the walking dead game season 4, you'll provide answers for Clementine and I'll take the role of L
uis...'
    0.C the user wants to request dialogue writing; specifically, the query focuses on generating specific conversations between characters. For exa
mple, 'Write dialogue from a scene from the animated teen "Jane", where 14 year old Jane, Jane's 14 year old girlfriend Sam, Jane's 14 year old friend an
d neighbour Aaron Ling and Aaron's 12 year old sister Molly-Rose Ling hanging out at school...'
    0.D the user wants to request content rewriting; specifically, the query asks for altering or reimagining existing content within a new context.
 For example, 'Can you rewrite 1972's Ben as a Tales From Crypt movie, it goes the same way but write the intro and outro of the Crypt Keeper?'

===============
1 the user wants assistance with writing or editing text; specifically, the query involves tasks like composing positive feedback or rewriting content. F
or example, 'Please re-write this as a positive feedback for Stelina, praising her.'.
    1.A the user wants to create new fictional content; specifically, the query involves generating original stories, scripts, or characters. For ex
ample, 'Write dialogue from a scene from the animated teen Jane, Jane's 14 year old girlfriend Sam and Jane's 14 year old friend
and neighbour Aaron Ling hanging out at school when Jane and Sam finds out Aaron has a crush.'
    1.B the user wants to request dialogue writing for fictional characters; specifically, the query requests the creation of conversation between c
haracters in a specific scenario. For example, 'Write dialogue from a scene from the animated teen "Jane"...'.
    1.C the user wants to request a short response or reply composition; specifically, the query involves crafting a concise message to send in a di
scussion. For example, 'give me a response to *smiles* Yes, I'm doing well. How about you?'.
    1.D the user wants to request content rewriting and optimization; specifically, the query involves improving the language, structure, or adding
references to existing text. For example, 'summarize this parts for method section in an article and rewrite it to improve and add some references'.

===============
2 the user wants to request feedback or evaluation; specifically, the query involves asking for re-writing or giving feedback on a piece of text. For exa
mple, 'Please re-write this as a positive feedback for Stelina, praising her.'
    2.A the user wants to request a concise and clear response for a given message; specifically, the query involves crafting a short and direct rep
ly for use in a discussion. For example, 'give me a response to Absolutely. Compassion, understanding, and respect...'.
    2.B the user wants to request for grammar and spelling correction; specifically, the query involves correcting grammatical and spelling errors i
n a given text. For example, '( there was an assistance in our village about Stationary and Hygiene's for the students of (Nasozai Markazai Lesa )...'.
    2.C the user wants to request an academic rewrite of text; specifically, the query involves rephrasing content to meet academic standards or exp
ressions. For example, '使下列文字符合学术表达方式 [text in Chinese].'
    2.D the user wants to request for summarization of information; specifically, the query involves condensing information into key points or a bri
ef summary. For example, 'In point form, simply summarize key insights on barriers to participate in procurement from the below text at an 8th-grade read
ing level and provide brief supporting quotes with ID numbers'.

===============
3 the user wants to request technical code/script writing; specifically, the query demands the creation or modification of a script or code. For example,
 '逐行解释以下代码 class MediaPipeFace:'.
    3.A the user wants to request data manipulation or computation; specifically, the query involves operations like calculations, data processing,
or data transformation. For example, 'how to get percentage of maxhealth to health in java'.
    3.B the user wants to request debugging assistance; specifically, the query involves asking for help to identify and fix errors or issues in a g
iven code. For example, '帮我看看这段代码有啥错误'.
    3.C the user wants to create or enhance UI/UX components within web development; specifically, the query requires designing, styling, or improvi
ng user interface elements. For example, 'can you do some nice font for it looking'?
    3.D the user wants to request the creation of a new code/script to perform a particular task; specifically, the query asks for writing code from
 scratch to achieve a specific functionality. For example, 'write a code for DES Encryption from scratch in python'.

===============
4 the user wants to format a description of an image generation prompt; specifically, the query involves structuring a prompt for use with image generati
on tools. For example, 'Here is a midjourney prompt formula...'
    4.A the user wants to create AI-generated content based on language and localization; specifically, the query involves creating prompts in diffe
rent languages while maintaining specific design instructions. For example, 'Imagine you're an expert Graphic Designer and have experience in 男孩 t-shir
t printing and also an expert Midjourney AI Generative prompt writer...'
    4.B the user wants to request for creative content generation; specifically, the query involves asking for extensive and imaginative description
s or stories about fictional characters or scenarios. For example, 'write a detailed fusion scenario of two anime characters.'
    4.C the user wants to request graphic design prompts for Midjourney AI; specifically, the query involves generating prompts for creating specifi
c types of images. For example, 'Imagine you're an expert Graphic Designer and have experience in 男孩 t-shirt printing and also an expert Midjourney AI
Generative prompt writer...'.
    4.D the user wants to request a comparison or fusion of different art styles or characters; specifically, the query involves merging distinct st
yles or characters into a single concept. For example, 'Freedom planet and Dragon ball: Goku betrayed and join the Freedom planet with beerus, whis, zeno
 and his guards. part 1'.

===============
5 the user wants to request information or answers regarding a specific topic; specifically, the query involves seeking knowledge or clarification. For e
xample, 'what are some open source operating systems.'
    5.A the user wants to ask for historical information on a specific topic; specifically, the query seeks details about events, practices, or item
s from the past. For example, 'Can you give me some summarized info on sugar cane 100 years ago?'
    5.B the user wants to ask for definitions or distinctions; specifically, the query seeks to clarify the meaning or differences between terms. Fo
r example, 'What is the difference between polygamy, polyandry, and polycules?'
    5.C the user wants to obtain scientific or factual information; specifically, the query is seeking accurate data or explanations relevant to aca
demic or scientific fields. For example, 'describe creature scientific name is felis achluochloros, evolved from feral creatures of felis catus, includin
g species name.'
    5.D the user wants to request technical issue resolution; specifically, the query involves troubleshooting or solving a problem with software or
 hardware. For example, 'how do I make a circle in CSS that sits on top of other elements and applies a gray scale filter to all of the elements intersec
ting its area?'
```

Figure 10: The full taxonomy that our algorithm generates to categorize user applications from a corpus of user chatbot queries.

# H    A Formal Description of Our Algorithm

A formal description of our algorithm can be seen in Algorithm H.

# I    Additional Details of Our paper

## I.1    Limitations of Our Framework and Our Experiments

As mentioned in Appendix G.3, our current system is slow, as its performance highly depends on
the LLMs to compute denotations correctly, it outputs semantically similar predicates that add little
information, and it is hard to control the predicates to satisfy certain properties (e.g. being a subset
of a parent category). Our experiments are limited since it assumes that the datasets and statistical
models we used are reflective of real world application. We made our best effort to gather text
clustering datasets that are commonly used in the literature (e.g. from [52, 38]) and defined models
that are plausibly useful for practitioners. Additionally, note that our evaluation on topic clustering
is more comprehensive than the prior work [52] by including two new datasets (Bills and Wiki);

**Algorithm 1** A formal description of our algorithm. **Argument**: $S$ is the number of steps we use to run our algorithm. **Output**: var_$\phi_{1\ldots K}$ is the list of $K$ predicates that we maintain, optimize, and return at the end of the algorithm. $\hat{w}$ are other parameters.

We first optimize the relaxed continuous predicates and discretize them (Line 3-10), and then iteratively refines the predicates (Line 11-21). During iterative refinement, we first find the least useful predicate $k$ (Line 12), then we only optimize the continuous representation of the least useful predicate while fixing other discrete predicates (Line 14 - 17); finally we discretize the $k^{\text{th}}$ predicate (Line 18, 19

---

1: **Arguments**: $S$
2: **Output**: var_$\phi_{1\ldots K}, \hat{w}$
3: $\tilde{\phi}_{1\ldots K} \leftarrow$ randomly sample $K$ embeddings $e_x$ to initialize $\tilde{\phi}$
4: **for** $t = 1$ to 10 **do**
5:    $\hat{w} \leftarrow \texttt{OptW}(\tilde{\phi}_{1\ldots K})$
6:    $\tilde{\phi}_{1\ldots K} \leftarrow \texttt{OptRelaxedPhi}(\hat{w})$
7: **end for**
8: **for** $k = 1$ to $K$ **do**
9:    var_$\phi_k \leftarrow \texttt{Discretize}(\tilde{\phi}_k)[0]$
10: **end for**
11: **for** $s = 1$ to $S$ **do**
12:    $k \leftarrow \text{argmax}_{k'}\texttt{Fitness}(\text{var\_}\phi_{-k'}, [\![\phi_{k'}]\!] = 0)$
13:    $\tilde{\phi}_k \leftarrow$ randomly sample an embedding$e_x$
14:    **for** $t = 1$ to 10 **do**
15:       $\hat{w} \leftarrow \texttt{OptW}(\text{var\_}\phi_{-k}, \tilde{\phi}_k)$
16:       $\tilde{\phi}_k \leftarrow \texttt{OptRelaxedPhi}(\text{var\_}\phi_{-k}, \hat{w})$
17:    **end for**
18:    $C_k \leftarrow \texttt{Discretize}(\tilde{\phi}_k)$
19:    var_$\phi_k \leftarrow \text{argmax}_{\phi' \in C_k \cup \{\text{var\_}\phi_k\}}\texttt{Fitness}(\text{var\_}\phi_{-k}, \phi_k = \phi')$
20:    $\hat{w} \leftarrow \texttt{OptW}(\text{var\_}\phi)$
21: **end for**
22: **return** var_$\phi, \hat{w}$
)

---

additionally, we used the exact same hyper-parameter across all clustering tasks, while [52] changed the hyper-parameters for different datasets.

## I.2   Cost of the Experiments

All of the experiments ran in Section 5 are estimed to cost at most 200 GPU hours on an A100 GPU with 40GB memory, and cost less than $20 of API credit for `gpt-3.5-turbo`. The experiments in Section 6 costs at most $50 of API inference credit, but we are constrained by rate limit.

## I.3   Licenses for Existing Datasets

[40, 58, 32, 23]) AG-News [58] has unknown license, the DB-Pedia dataset is released under Creative Commons Attribution Share Alike 3.0, the NYT dataset is distributed by LDC under the LDC's generic non-member license, the Bills dataset [23] are considered public domain works, and the Wiki dataset is licensed under CC BY-SA 4.0.

## I.4   License for the Assets Provided by Our Paper

Our code will be shared under CC BY-SA 4.0.

## I.5   Broader Impacts

This paper presents work whose goal is to advance the field of Machine Learning. Our framework could potentially make machine learning systems more explainable, thus making them safer, more trustworthy and easily auditable. On the other hand, however, the learned predicates only reflect

correlation rather than causations learned from data, and hence requires careful interpretation. Given that the performance of our model-agnostic method is still far from perfect and it is unclear how human users would use them in real world applications, the algorithm presented in this paper should only be used for research and not deployed in practice.

