# OpenReview forum: "Explaining Datasets in Words: Statistical Models with Natural Language Parameters"
_NeurIPS.cc/2024/Conference — NeurIPS 2024 poster_

### Official Review · Reviewer_krnw · 2024-07-14

**Soundness:** 3
**Presentation:** 3
**Contribution:** 3
**Rating:** 6
**Confidence:** 3

**Summary:**

The authors propose a general framework to generate explanations of different text datasets, by parameterizing the data distribution with textual predicates. The textual predicates, along with their weights, can be viewed as an explanation of the data. Three different tasks are used as examples: clustering, time-series modeling, and multi-class classification. To learn the textual predicates and their associated weights, the authors first iteratively learn the predicates in the embedding space along with the weights, and then use a pre-trained language model to discretize the predicate embeddings into textual space. Experiments are conducted on three datasets that have ground-truth predicates. A qualitative experiment with user-chatgpt dialogues is also presented.

**Strengths:**

1. The proposed framework nicely generalizes from the classical Bayesian text analysis methods and makes them more relevant/useful in the LLM era.

2. Comprehensive experiments are performed on four different datasets/tasks.

3. The paper is well-written and easy to follow.

**Weaknesses:**

1. Missing pure LLM baselines. It is fairly common nowadays that LLMs are directly used to generate explanations given the data, which can be applied to any dataset and any task. More evidence is needed to show the advantages of the proposed method compared to direct prompting. For example, for the clustering task, what if you perform clustering first, and then prompt GPT4 to directly explain each cluster?

2. The explanations generated are mostly topic descriptions, and the datasets used for qualitative results are quite simple. Can the proposed method generate more complex descriptions for more difficult datasets? For example, for a STEM problem-solving dataset (e.g. MATH), can the proposed method explain the dataset by the theorem used/subarea of the subject/similar solving strategy?

In summary, I like the way the proposed method combines the classical Bayesian method with LLM prompting, but I'm not totally convinced that the proposed method is significant as there is no direct comparison with LLM prompting, or evaluation on harder datasets. I'm willing to raise my score if my concerns are properly addressed.

**Questions:**

See weaknesses.

**Limitations:**

Yes.

---

> ### Author Rebuttal · Authors · 2024-08-02
>
> Thanks for your review and recommended experiments!
>
> ## Summary of Our Response
>
> ### Re: pure LLM prompting baseline.
>
> - This baseline is similar to our no-refine and no-relax baseline, and we showed that both refinement and relaxation were helpful (Takeaways 1 & 2).
> - We **directly ran the naïve prompting experiment you recommended** (Tables 1 & 2 in the author response pdf), and **our method is significantly better**.
>
> ### Re: clustering on more complicated datasets (e.g. MATH).
>
> As you recommended, we ran our clustering model on the MATH dataset.
> - Our clustering model **recovered all the 5 subareas predefined in the MATH dataset**
> - Our model was able to cluster solutions based on the strategies to solve the problems
>
> Below is our more detailed response.
>
> ## Re: Missing Pure LLM baseline
>
> Our paper implicitly compared our method to this baseline. Effectively, no-refinement + no-relaxation ~=pure LLM prompting. Without relaxation, the language model would propose predicates only based on the text samples, and we will directly output these predicates under no-refinement. Our paper has shown that both refinement and relaxation are helpful (Takeaways 1 and 2 in Section 5), implying that our approach will outperform the naïve prompting baseline.
>
> Still, we completely agree that including the pure LLM prompting baseline can better highlight the strength of our algorithm, and indeed we find that **our algorithm significantly outperforms this baseline**.
>
> Specifically, we directly prompted the language model to generate a list of natural language predicates. Here is a part of our prompt for clustering: “We want to come up with some features (expressed in natural language predicate) such that they can be used for clustering”. To control for the randomness in random text samples in the language model context, we ran this baseline approach 5 times and report the average performance in Table 1 and 2 in the author-response pdf.
>
> Across all datasets, the performance of prompting LLM lags significantly behind our approach, indicating the necessity of a sophisticated approach. Additionally, the max of 5 seeds performance (F1-score) for clustering/time-series/classification is 0.36/0.45/0.58 on average, which still significantly lags behind our approach (0.64/0.64/0.73). This implies that our method is robustly better than the naïve, pure LLM baseline.
>
>
> ## Applying our method to more complicated tasks (e.g. sub-area/solution strategy on the MATH dataset)
>
>
> Our evaluation focused on topic-based datasets since it is a standard practice in the clustering literature [1,2] and these datasets have ground truth. However, indeed our method is not limited to topic clustering. **As you have recommended, we ran our clustering model on the MATH dataset**. We will first present the results, and then describe the experimental details.
>
> When clustering problems based on subareas, **our method recovered all of the 5 subareas in the MATH dataset as defined by the original authors**. The MATH dataset was released under five pre-defined categories (after merging different difficulty levels of algebra):
>
> ```[Algebra, counting_and_probability, geometry, number_theory, precalculus]```
>
> Here is the output of our clustering model:
> ```
> - involves algebraic manipulation;
> - involves probability or combinatorics
> - requires geometric reasoning;
> - pertains to number theory;
> - involves calculus or limits;
> ```
> We then clustered the geometry solutions based on their strategies. While we do not have ground truth clusters to compare against, we qualitatively examined a few examples for each cluster and most of the samples satisfy the corresponding cluster descriptions. Here is our outputs
> ```
> - employs similarity or congruence in triangles;
> - uses algebraic manipulation to solve for unknowns;
> - involves area or volume calculation;
> - uses geometric diagrams or visuals;
> ```
>
> To conclude, we directly ran the experiments you recommended, showing that our method significantly outperforms the naïve prompting baseline and can be applied to more challenging tasks beyond topic clustering.
>
> **Experimental Details and Results:**
>
> When clustering the problem descriptions based on subareas, we prompt the LM “I want to cluster these math problems based on the type of skills required to solve them.” to discretize. When clustering the solutions based on the strategies, we used the following prompt: "I want to cluster these math solutions based on the type of strategies used to solve them."
>
> [1] Goal-driven explainable clustering via language descriptions
>
> [2] TopicGPT: A Prompt-based Topic Modeling Framework

---

> ### Author Response · Authors · 2024-08-10
> **Looking forward to your reply**
>
> We have directly run the experiments mentioned in your review. Please feel free to let us know if you have any questions. Thanks a lot!!

---

> > ### Comment · Reviewer_krnw · 2024-08-12
> >
> > Thank you for the rebuttal and the new experiments. I'll increase my score in light of adding these new results to the paper.

---

> > > ### Author Response · Authors · 2024-08-12
> > > **Thanks for your support**
> > >
> > > Thanks for reading our response and supporting our work!

---

### Official Review · Reviewer_ZMnS · 2024-07-16

**Soundness:** 3
**Presentation:** 3
**Contribution:** 2
**Rating:** 5
**Confidence:** 4

**Summary:**

This paper proposes to use "natural language predicates" to explain text datasets. Authors develop a model-agnostic algorithm that optimizes continuous relaxations of predicate parameters with gradient descent and discretizes them by prompting language models. With this method, the authors can characterize the dataset with a distribution over predicates. The authors apply it to taxonomize real-world user chat dialogs and show how the text data's features can evolve across time, demonstrating the potential of the method in understanding text data.

**Strengths:**

1. The paper proposes a new method to explain text datasets. There are a few new concepts in the paper, such as "Predicate-Conditioned Distribution" and the corresponding optimization algorithm that involve both continuous and discrete optimization.

2. A practical application of the method is demonstrated on real-world user chat dialogs.

3. The problem is well-formulated and backed by enough mathematical and empirical evidence.

**Weaknesses:**

1. The motivation of the paper is somewhat unclear. It's not well explained in the Introduction section why people want to "Explain text datasets". Is it for controlling the dataset quality for pretraining? Or tailor the domain of dataset for finetuning LM? Although the authors showcase the method on a dataset of real-world user-ChatGPT dialogues, the results look arbitrary and not very insightful. Only "Taxonomizing User Applications via Clustering" and "Characterizing Temporal Trends via Time Series Modeling" are still not fully demonstrating the usefulness of the method.

2. The method should be compared with more baseline methods. It's unknown what's the benefit of predicate-conditioned distribution over other methods. The authors should also explain why the "predicate" is the key to explain text datasets, instead of other factors such as just verbs, nouns, text representations, etc.

3. The datasets being tested are relatively small and not showing the scalability of the method.

**Questions:**

1. Can you provide more insights on the motivation of the paper? Why do people want to explain text datasets? What are the potential applications of the method apart from the two demonstrated in the paper?

2. What is the benefit of using "predicate" to explain text datasets? How is the predicate better than other forms of explanations?

**Limitations:**

Limitations are being discussed in the paper.

---

> ### Author Rebuttal · Authors · 2024-08-02
>
> Thanks for your thoughtful review. We will
> - Clarify our motivation to explain datasets
> - We compared our method to the approach you have recommended (representing clusters with verbs/nouns). **We find that natural language predicates can provide much better explanations than individual words**.
> - We ran the experiment above on a dataset with 12.5K examples, mitigating your concern that our method only works on small datasets.
>
> ## The motivation for explaining datasets?
>
> There are many existing machine learning and data (social) science applications for explaining datasets.
>
> In the social science/data science domain,
> - [1] categorizes online discussions about political topics by generating natural language descriptions.
> - [6] clustered datasets to mine arguments in debates [2] and customer feedbacks in product reviews [9]
>
> Our framework can also be potentially applied to
> - understand what makes human thinks that a text is machine-written (with our classification model) [3]
> - analyze Google search trend similar to [4]
>
> Prior works [5] and [6] include comprehensive overviews of how clustering and classification models with natural language parameters are useful, separately. Our work contributes by connecting these separate models and creating a unified formalization and optimization framework.
>
> ## The significance of taxonomizing LLM applications.
>
> This is a growing need from the community that hosts Chatbots for users.
> - Industry: LLM companies are building internal prototype systems like this to understand how their LLM is used with real user data.
> - Academia: The ChatbotArena used naïve method to cluster user queries (Figure 3 of [7]) and prompt LLM to summarize the clusters, indicating that this is a realistic application that some machine learning researchers care about. They are currently trying to taxonomize these queries to create more fine-grained Elo ratings.
>
> ## Why natural language predicates, but not other representations like verbs or adjectives?
>
> In L266 and Appendix Table 4, we show that the word-based topic model significantly underperforms our method.
>
> Prior work [8] has also found that it is hard to interpret meaning from a list of representative words since they might not be semantically coherent, and [6] shows that natural language predicates are more explainable than lists of words for humans.
>
> We illustrate this by including an additional experiment suggested by the reviewer krnw: clustering the MATH dataset based on their subareas. We will first present the results and then describe the experimental details. Overall, **we find that significant guesswork is needed to interpret the meaning of each word-based cluster (e.g. unclear what cluster 1 represents), while the predicates generated by our algorithm are directly explainable.**
>
> Outputs of our model:
> ```
> - involves algebraic manipulation;
> - involves probability or combinatorics
> - requires geometric reasoning;
> - pertains to number theory;
> - involves calculus or limits;
> ```
>
> Word-based cluster representation (presenting two based on space limit):
> ```
> cluster 0:
> > 'ADJ': ['shortest', 'cartesian', 'parametric', 'polar', 'correct'],
>
> > 'ADV': ['shortest', 'numerically', 'downward', 'upward', 'meanwhile'],
>
> > 'NOUN': ['areas', 'option', 'foci', 'endpoints', 'perimeter'],
>
> > 'VERB': ['lie', 'enclosed', 'corresponds', 'vertices', 'describes']},
>
> cluster 1:
>
> > {'ADJ': ['decimal', 'base', 'sum_', 'residue', 'prod_'],
>
> > 'ADV': ['90', 'recursively', 'mentally', 'dots', 'left'],
>
> > 'NOUN': ['operation', 'sum_', 'denominator', 'dbinom', 'div'],
>
> > 'VERB': ['evaluate','simplify', 'rationalize','calculate', 'terminating']},
> ```
>
> ## Experimental details:
>
> We run the experiment suggested by reviewer krnw: clustering the MATH dataset based on their subareas. We directly compared our clustering algorithm to the approach you recommended: concretely, we first cluster the problem description based on K-means; then for each cluster, we filter out the nouns/verbs/adjectives, learn a unigram regression model to predict whether a sample comes from a cluster, and present the unigrams with the highest coefficient.
>
> [1] Opportunities and Risks of LLMs for Scalable Deliberation with Polis
>
> [2] DebateSum: A large-scale argument mining and summarization
>
> [3] Human heuristics for AI-generated language are flawed
>
> [4] Google Trends
>
> [5] Goal-Driven Explainable Clustering via Language Descriptions
>
> [6] Goal Driven Discovery of Distributional Differences via Language Descriptions
>
> [7] Chatbot Arena: An Open Platform for Evaluating LLMs by Human Preference
>
> [8] Reading Tea Leaves: How Humans Interpret Topic Models
>
> [9] Huggingface dataset: ​​McAuley-Lab/Amazon-Reviews-2023

---

> > ### Author Response · Authors · 2024-08-12
> > **Looking forward to your comment**
> >
> > We have run the experiments to address your reviews. Please feel free to let us know if you have any questions. Thanks a lot!!

---

### Official Review · Reviewer_fF31 · 2024-07-20

**Soundness:** 4
**Presentation:** 4
**Contribution:** 4
**Rating:** 7
**Confidence:** 4

**Summary:**

The paper introduces a a framework for a family of models parameterized by natural language predicates. This models allow for a language-based interpretation of text distribution. Such framework can easily instatiate models include clustering, time-series, and classification models. The authors develop a model-agnostic algorithm that optimizes continuous relaxations of predicate parameters with gradient descent and then discretizes them by prompting language models (LM). This approach is demonstrated to be effective in taxonomizing news dataset as well user chat dialogues while also characterizing their evolution over time.

**Strengths:**

I find the paper to have strengths along several axes:

- Novelty: The general framework is, to the best of my knowledge a novel approach to describe text datasets in a way that is interpretable and scalable.

- Model-Agnostic Algorithm: The algorithm developed to optimize the models proposes is versatile and can be applied to various types datasets under different approaches, namely classification, clustering and time-series analysis.

- Extensive experiments and interesting ablations: The authors demonstrated the practical utility of the approach by applying it to real-world datasets (user-LLM chats), as well as news data for comparison purposes.

- The results show improvements over the baseline and when this is not the case Bills and NYT dataset for clustering in table 2, the difference is either minimal or compensated with the generality of the proposed method.

Overall the paper is well written and easy to follow. The results well presented and support the claim made in the paper. The limitation as well as weaknesses of the work are presented and discussed.

**Weaknesses:**

I did not find major weaknesses in this paper.

- The main one would be in the addition computational and money costs for the proposed method compared to other classical approaches. However, in my opinion, the additional costs are counter balanced by performance and generality of the method.

- Another weakness is the use of a single text embedding model. I do not think this addition experiment is necessary for accepting this paper, but I do think that testing the robustness of this method with various text embedding would make the presented claims stronger.

- Additionally, given the experiments are averages across several seeds, I would report the variance of the results.

----

MINOR

- Please adjust the citation format, in the text there's a number-based format but there are not numbers in the references

- Very minor: then I tried to read table table 1 (pag 6), I was confused about what metric surface was. This was explained later in page 7. I would add a brief explanation of surface in the caption of the table, make a reference to the section when surface is explained or move the table closer the experiments section.

- Typo in caption of table 2: relaxiation -> relaxation

- line 277 general purpose -> general-purpose

- line 648: you refer section G.1 from section G.1, did you mean section E.3?

---

Recomendation

Overall I consider this a strong a paper and I vote in favor of its inclusion to NeurIPS

**Questions:**

- How are you selecting language predicates lengths? Given the NLG component in the LLM call, can predicate length evolve over time? Have you done an analysis of such change in your optimization process?

- When you mentioned ranking by correlation scores: the correlation is computed between $[[\phi]]$ and the dot product between a predicate relaxation and a sample embedding. Are you correlating continuous vs discrete values? Is $[[\phi]]$ a binary value? Could you please clarify your correlations here?

- In the no-refine case, am I reading the results correctly If I understood that you select random predicates and discretize after one full round of optimization (of $w$ and $\phi$ them once?

- In the no-relax case, you are still discretizing using a language model, is that correct? Would this just mean that the optimization process would take longer given that you are skipping the correlation-based selection criterion?

**Limitations:**

Yes, in section 6, E.3 and G.1.

---

> ### Author Rebuttal · Authors · 2024-08-02
>
> Thanks for appreciating the strength of our paper! We will address each of your questions below.
>
> ### Checking whether our conclusion generalizes to other embedding models
>
> We report the performance of another embedding model, all-mpnet-base-v2, and report the results in Table 1 & 2 in the author response pdf. In all but 2 datasets, using this embedding model outperforms not using an embedding-based model (one-hot), indicating the robustness of our approach.
>
> ### Reporting the cost of fitting our model
>
> We report the cost of our experiments in Appendix G.2. If we use a local validator of google/flan-t5-xl (1.5B), fitting each model reported in Table 2 and 3 takes around one hour, and clustering a corpus with 1M tokens with K=8 clusters takes around $9 using the latest GPT-4o-mini model. Indeed, it is currently much more expensive than traditional methods like K-means clustering (e.g. taking around 1 minutes to cluster and find salient words). As you have mentioned, the additional costs are counterbalanced by performance and generality of the method. Finally, our method is likely to become cheaper in the future – the cost for API calls given the same capability level has decreased drastically (e.g.~10^4) since GPT-3 has initially released, and we expect the cost to continue going down.
>
> ### Reporting the variance of the experiments
>
> We will report the standard deviation of our method in our updated version.
>
> ### How is the correlation between the continuous predicate and the discrete predicate calculated?
>
> The continuous predicate is a continuous function that maps from a text sample to a real-value, and the denotation of the discrete predicate is a binary function that maps from a text sample to an integer of 0/1. For models where $w$ is only non-negative (e.g. $\infty$), we compute the pearson-r correlation between the two functions across all text samples; for models where $w$ can be negative, we compute the absolute value of the pearson-r correlation.
>
> ### Clarification in the no-refine case
>
> The no-refine algorithm is described in Section 4.3, the initialization stage. We would:
> - Alternatively optimize two variables: 1) $w$, and 2) the continuous representation of ALL the predicates (instead of random ones)
> - Discretize each continuous representation and find one most correlated predicate
>
> The continuous representation of ALL the predicates are randomly initialized by randomly selecting K embeddings of the text samples (K-means++initialization[1] ).
>
> ### Does the predicate length evolve throughout time?
>
> Averaged across all tasks reported in Table 2-3, the average predicate length after the first discretization is 5.99, while it is 6.03 at the end. Ex-ante it is plausible that the predicates will become longer as LLM “refines” their predicates by making them more precise; however, empirically the loss usually decreases because the LLM discovers a completely new feature (e.g. cluster) that has little correlation with the feature before the update.
>
> ### Clarification on the no-relax case. Do you still use the language model? Do they take longer to converge?
>
> Yes, we are still using the language model; however, without any scoring on the text samples, this is similar to proposing random predicates conditioned on all the text samples, in an uninformed manner. We average across the loss curve across all tasks and plot it in Figure 1 our author response pdf. We find that the speed of convergence is significantly faster with relaxation.
>
> Thanks a lot for your other feedback on writing! These are really useful feedback and we plan to incorporate them in our updated version.
>
> [1] k-means++: the advantages of careful seeding

---

> ### Comment · Reviewer_fF31 · 2024-08-08
> **Response to rebuttal**
>
> Thanks a lot for your work on the rebuttal.
>
> You response confirms my initial idea about the high-quality and solid contribution of your work.
>
> I confirm my score and vote for the inclusion of the paper at the conference

---

> > ### Author Response · Authors · 2024-08-08
> > **Thanks for supporting our work**
> >
> > Thanks for reading our response and supporting our work!

---

### Author Rebuttal · Authors · 2024-08-02

Thanks to all the reviewers for the feedback! We are glad that overall the reviewers like our formulation, method, and writing. In this response, we supplement new experiments to strengthen our paper.

- Comparison against naive prompting (reviewer krnw): while our paper has implicitly compared to this baseline, we followed the recommendation to directly run this experiment. **We find that our method is significantly better than this naïve prompting baseline**.
- Clustering on the MATH dataset (reviewer krnw, ZMns): we directly ran the experiment suggested by krnw, and **our method uncovers all the 5 ground truth subarea in MATH**. Additionally, we compared our approach to the verb/noun-based method suggested by reviewer ZMns and found that our approach is more explainable.
- Generalization to other embedding models (reviewer fF31): we ran our experiments on another embedding model (all-mpnet-base-v2) and found that it also significantly outperforms the one-hot-embedding baseline, further validating the importance of using informative text embeddings.

We have directly addressed most of the concerns raised by the reviewers. If you have any further questions, feel free to let us know!

---

### Decision · Program_Chairs · 2024-09-25

**Decision:**

Accept (poster)

**Comment:**

This paper proposes a general framework to generate explanations of text datasets, by parametrizing the data distribution with textual predicates. The textual predicates, along with their weights, can be viewed as an explanation of the data. To learn the textual predicates and their associated weights, they iteratively learn the predicates in the embedding space along with the weights, and then use a pre-trained language model to discretize the predicate embeddings into textual space.

Most reviewers agree that this is an interesting and novel contribution. Most of the concerns seem to have been addressed convincingly in the author rebuttal and in the additional experiments. I recommend acceptance.